# Diffusion- and Chemometric-Based Separation of Complex Electrochemical Signals That Originated from Multiple Redox-Active Molecules

**DOI:** 10.3390/polym14040717

**Published:** 2022-02-13

**Authors:** Stav Biton Hayun, Rajendra P. Shukla, Hadar Ben-Yoav

**Affiliations:** Nanobioelectronics Laboratory (NBEL), Department of Biomedical Engineering, Ilse Katz Institute of Nanoscale Science and Technology, Zlotowski Center for Neuroscience, Ben-Gurion University of the Negev, Beer-Sheva 8410501, Israel; stavbi@post.bgu.ac.il (S.B.H.); rajendra@post.bgu.ac.il (R.P.S.)

**Keywords:** chitosan, electroanalysis, chemometrics, multi-sensor arrays, partial least squares, electrochemical sensors, ascorbic acid, clozapine, uric acid, homocysteine

## Abstract

In situ analysis of multiple biomarkers in the body provides better diagnosis and enables personalized health management. Since many of these biomarkers are redox-active, electrochemical sensors have shown promising analytical capabilities to measure multiple redox-active molecules. However, the analytical performance of electrochemical sensors rapidly decreases in the presence of multicomponent biofluids due to their limited ability to separate overlapping electrochemical signals generated by multiple molecules. Here we report a novel approach to use charged chitosan-modified electrodes to alter the diffusion of ascorbic acid, clozapine, L-homocysteine, and uric acid—test molecules with various molecular charges and molecular weights. Moreover, we present a complementary approach to use chemometrics to decipher the complex set of overlapping signals generated from a mixture of differentially charged redox molecules. The partial least square regression model predicted three out of four redox-active molecules with root mean square error, Pearson correlation coefficient, and R-squared values of 125 µM, 0.947, and 0.894; 51.8 µM, 0.877, and 0.753; 55.7 µM, 0.903, and 0.809, respectively. By further enhancing our understanding of the diffusion of redox-active molecules in chitosan, the in-situ separation of multiple molecules can be enabled, which will be used to establish guidelines for the effective separation of biomarkers.

## 1. Introduction

Simultaneous and continuous monitoring of multiple redox-active biological molecules in biofluids can provide better and more information about physiological processes in the body. Such enhanced analysis can lead to improved diagnosis and treatment of diseases. For example, changes in the redox balance (the antioxidants to pro-oxidants redox molecules ratio) in the blood indicate an oxidative stress state during cancer and other metabolic disorders [1,2,3]. Current methods for the simultaneous measurement of multiple redox-active molecules in biofluids, such as gas chromatography, high-performance liquid chromatography, and capillary electrophoresis, are time-consuming, and they rely on the pre-separation of the molecules. Despite the improved sensitivity of these methods, the laborious, long, and expensive preparation steps prevent rapid monitoring of the molecules in the biofluids [3,4,5,6,7].

Electrochemical sensors have attracted much attention to overcome these problems due to their fast response, small size, simple operation, low-cost, and real-time monitoring capabilities for continuous and simultaneous detection of molecules in situ. Moreover, these translational approaches are perfectly suited for analyzing redox molecules due to the electrochemically active nature of the molecules. They enable generating quantitative electronic signals due to their oxidation or reduction reactions at the electrode surface. However, current electrochemical sensors suffer from their limited selectivity in the presence of biofluids due to the overlapping interfering electrochemical signals generated by other redox-active molecules with a similar redox potential [8,9,10,11,12,13,14,15,16,17].

A novel class of electrochemical sensors, namely, voltammetric electronic tongue sensors, showed the ability to separate multiple redox molecules simultaneously. This type of sensor consists of an array of semi-selective electrodes, which allows quantifying multiple analytes simultaneously using data processing algorithms such as principal component analysis, artificial neural networks, and partial least squares [18,19]. Electronic tongue arrays include several different electrodes made of different materials [20,21] or modified with different coatings [22,23]. Despite the high contribution of voltammetric electronic tongue sensors to environmental, food, and pharmaceutical applications regarding biomedical analysis in biofluids, voltammetric electronic tongue sensors suffer from limited sensitivity due to the highly complex nature of a sample containing multiple redox analytes in the presence of other redox molecules with similar redox potentials that generate overlapping electrochemical signals.

Here we propose a novel approach to “process” in situ differently charged redox molecules with similar redox potentials by altering their diffusion rate (Figure 1). For this purpose, we utilized the biopolymer chitosan—a positively charged and a pH-responsive hydrogel—to modify the electrode surface [24,25,26]. We selectively modify electrodes with different chitosan film properties (i.e., film thickness and density [24,27]) to create complex porous structures [28] of various permeabilities [29,30,31]. Our working hypothesis is that by varying the parameters of the chitosan film, its thickness and pore sizes can be tuned to alter the diffusion coefficients of redox molecules through electrostatic interactions and by varying the diffusion paths, which will influence the underlying electrochemical signal [32]. Here we demonstrated a strong dependency of the effective diffusion coefficient on the chitosan thickness—for a 54.7 nm-thick chitosan film, using negatively charged ascorbic acid (AA). This resulted in an effective diffusion coefficient that was higher than of the neutrally charged L-homocysteine (HCy) and of the positively charged clozapine (CLZ). In addition, we showed the ability to use the chemometric model’s partial least squares regression (PLSR) to differentiate between overlapping electrochemical signals generated from a multicomponent mixture containing four redox-active molecules (AA, uric acid [UA], HCy, and CLZ) and to successfully quantify three of the molecules (AA, UA, and HCy. The calculated Pearson’s correlation coefficients [PCC] were 0.947, 0.877, and 0.903, respectively).

## 2. Materials and Methods

### 2.1. Chemical and Reagents

Chitosan (>85%, catalog number: 448869, Sigma-Aldrich, Ltd., Rehovot, Israel), clozapine (catalog number: C6302, Sigma-Aldrich, Ltd.), uric acid (catalog number: 01935, Chem-Impex, Wood Dale, IL, USA), ascorbic acid (catalog number: BIA0602, Apollo Scientific Ltd., Cheshire, UK), L-homocysteine (catalog number: BIB6065, Apollo Scientific, Ltd.), di-sodium hydrogen phosphate dihydrate (catalog number: 1.06580.1000, Merck, Darmstadt, Germany), sodium dihydrogen phosphate dihydrate (catalog number: 1.06342.0250, Merck), sodium chloride (catalog number: 1.06404.1000, Merck), potassium hexacyanoferrate (II) trihydrate (‘Ferro’, catalog number: 1.04984.0100, Merck), potassium hexacyanoferrate (III) (‘Ferri’, catalog number: 1.04973.0100, Merck), acetone (catalog number: 1900900178, Romical, Be’er Sheva, Israel), 2-propanol (catalog number: 001626052100, Bio-Lab, Ltd., Jerusalem, Israel), methanol (001368052100, Bio-Lab, Ltd.) and hydrochloric acid 32% (catalog number: 000846050100, Bio-Lab, Ltd.) were used without further purification. Ultra-pure water (>18 MΩ) was obtained from a Super Q water system (Millipore system, Merck). A concentrated chitosan solution (1.8%, pH 5.5) was prepared by dissolving chitosan flakes in 2 mol L^−1^ HCl to reach a final pH of 5–6. Then, the concentrated chitosan solution was diluted with Mili-Q water to a 1% chitosan solution. All solutions were prepared on the day of measurement.

### 2.2. Fabrication of the Gold Electrodes

A conventional microfabrication process was used to pattern four gold disk electrodes (2 mm in diameter) onto a Si/SiO_2_ substrate. The microfabrication process is based on photolithography and wet etching techniques (Figure 2); a 20 nm-thick layer of Ti was evaporated onto a 4-inch silicon wafer (p-type, orientation: <100>, resistivity: 10–20 ohm-cm, oxide thickness: 500 nm, single side polished, prime grade; University Wafer, Inc., South Boston, MA, USA) and was followed by another 200 nm-thick layer of gold (E-gun deposition system, VST Service, Ltd., Petah Tikva, Israel). The next step was to spin (80RC Delta, Universal Spin-coating system, SUSS MicroTec, 2200 rpm for 12 s) on the wafer the positive photoresist Ti-xlift (MicroChemicals, Ulm, Germany), followed by a soft bake (using a contact hot plate at 110 °C for 2.5 min). The coated wafer was exposed to our designed transparency mask (CAD/Art Services, Bandon, OR, USA) at a UV light flux of 7.6 mW cm^−2^ for 65 s (Karl Suss Mask aligner MA6 system, SUSS MicroTec, Garching, Germany). The photoresist was then developed in an AZ 726 MIF developer (DEAA174517, Merck) for 8 min. The developed wafer was rinsed in deionized water for 5 min and was dried with nitrogen gas. An Au wet etching step (10 s long) followed by a Ti wet etching step (6 s long) was used to define the electrodes’ patterns. The Au etching solution (4 g potassium iodide and 1 g iodine dissolved in 40 mL deionized water) and the Ti etching (Transene solution) were used without further purification. The wafer was rinsed in deionized water for 1 min between both etching processes and after the Ti etching step. Finally, the remaining photoresist was removed by using acetone.

To achieve a chemically stable electrochemical chamber, we used photolithography to pattern a SU-8 (SU08-3050, MicroChemicals) negative photoresist that was further hard-baked. The SU-8 photoresist was spanned (80RC Delta, Universal Spin-coating system, SUSS MicroTec, 3000 rpm for 30 s) onto the Si/SiO_2_ wafer, followed by a soft bake (using a contact hot plate at 95 °C for 15 min). The coated wafer was exposed to our designed transparency mask (CAD/Art Services) at a UV light flux of 7.6 mW cm^−2^ for 50 s (using a Karl Suss Mask aligner MA6 system, SUSS MicroTec), followed by a Post-Exposer Bake (using a contact hot plate at 95 °C for 5 min). The wafer was cooled down to room temperature, and then the photoresist was developed in a PGMA ERB developer (DEA165166, Merck) for 8 min and rinsed slowly. The developed wafer was rinsed in deionized water for 5 min and dried with nitrogen gas. After development, the wafer was washed in an IPA solution for 10 s. Then, a hard bake step (using a contact hot plate at 150 °C for 5 min) was followed by an Oxygen Plasma cleaning step (30 W, 500 mtorr, 2 min, 3 sccm). Finally, the wafer was diced into electrochemical testing chips by using a Dicer saw (Dicer ADT-7100, ADT). Before the electrochemical testing, cleaning was performed using acetone, methanol, and isopropanol solution, followed by rinsing with deionized water and drying with a nitrogen gun.

### 2.3. Chitosan Electrodeposition onto the Electrodes

A chitosan biopolymer was electrodeposited onto the bare Au microfabricated electrode. A chitosan solution with a density of 1% and a commercial Pt wire counter electrode (CH115, CH Instruments) was used for the electrodeposition process (Chronopotentiometry technique: a current density of 4 Am^−2^, I = −15.7 µA. The time was changed from 30 s to 180 s; VSP-300 potentiostat, Bio-Logic, Ltd., Seyssinet-Pariset, France). After the modification, the chip was dipped in 10 mM PBS for 1 min to remove any unbound chitosan.

### 2.4. Electrode Surface Characterization

The chip was first dried using a nitrogen gun for profile measurements under dry conditions. The thickness of the electrodeposited chitosan layer was measured by using a profilometer (Dektak-8, Veeco, Ltd., Tucson, AZ, USA). The measurements were taken at three different locations on the electrode, and the thickness’s mean and standard deviation was calculated.

### 2.5. Electrode Electrochemical Characterization

Prior to the electrodeposition of the chitosan onto the Au electrode, the electrochemical activity of the microfabricated electrode was examined in a 5 mM ferrocyanide/ ferricyanide (‘Ferro/Ferri’) testing solution using a cyclic voltammetry (CV) technique (vertex potential E_1_ = E_i_ = −0.1 V vs. Ref., vertex potential E_2_ = 0.65 V vs. Ref., scan rate = 0.050 V s^−1^, the repeat number of cycles n_c_ = 5; VSP-300 potentiostat, Bio-Logic, Ltd.). Electrochemical analysis of the redox molecules (Ferri, AA, HCy, and CLZ) was performed by assembling a multi-working electrode electrochemical chamber comprising 24 microfabricated working electrodes, a commercial Pt wire as the counter electrode (Platinum counter electrode 23 cm, 012961; ALS Co., Ltd., Tokyo, Japan), and a metal wire coated with Ag/AgCl ink as the reference electrode (Ag/AgCl ink for the reference electrode, BAS, Inc., Japan). Measurements were performed by applying a differential pulse voltammetry (DPV) electrochemical technique (E start −0.1 V vs. Ref., E end 0.7 V vs. Ref., Pulse time 1 ms, Pulse amplitude 55 mV, E step 1 mV, scan rate 10 mV s^−1^, Equilibration time 10 s; MultiWE32 and Ivium CompactStat potentiostat, Ivium, Ltd., Eindhoven, Netherlands) to all working electrodes simultaneously. Electrochemical signals of all the molecules were measured using solutions prepared in 10 mM PBS with various concentrations of Ferri (1, 2, 3, 4, and 5 mM), AA (100, 200, 500, 700, 1000, and 5000 µM), HCy (150, 300, 400, 500, and 600 µM) and CLZ (1, 2.5, 3.75, 5, and 10 µM). The concentrations were chosen based on their physiological levels in serum. Furthermore, Ferri concentrations were chosen based on standard levels used for electrochemical analysis [33,34]. Prior to each measurement, the electrodes were rinsed with 10 mM PBS. The simultaneous measurements with the multi-electrode electrochemical chamber were performed with bare and modified (30, 60, 90, 120, and 180 s chitosan electrodeposition durations) electrodes. All electrochemical measurements were performed at room temperature.

## 3. Results

### 3.1. Electrochemical Characterization of the Microfabricated Gold Electrodes

We characterized the electrochemical activity of the microfabricated Au electrodes and compared the results to that of a commercial Au electrode with a similar surface area. The recorded CV in the presence of the commonly used redox couple Ferro/Ferri revealed a similar Nernstian behavior (Figure 3A). Moreover, we measured similar peak currents and potentials for the oxidation (36.1 µA and 0.29 V for the microfabricated electrode and 37 µA and 0.27 V for the commercial electrode) and reduction (−36.5 µA and 0.18 V for the microfabricated electrode and −38.2 µA and 0.18 V for the commercial electrode) reactions.

We further compared the electroanalytical activity of the electrodes by calculating the effective diffusion coefficient (*D_eff_*) of Ferri. A calibration curve was plotted between different Ferri concentrations and the peak current (*i_p_*) values extracted from voltammograms recorded using a DPV technique (Figure 3B). A linear regression analysis of the calibration curve (Figure 3C) resulted in a linear relationship (Equation (1); R^2^ = 0.99):(1)$$i_{p}=\left[7.24\;\pm \;0.34\right]\left[Ferri\right]+\left[4.10\;\pm \;0.52\right],$$

We used the calculated slope of the calibration curve to calculate *D_eff_* (Equation (2) [32].
(2)$$i_{p}=\frac{nFAD^{0.5}\left(1-e^{\frac{nF\Delta E}{2RT}}\right)}{\pi^{0.5}t^{0.5}\left(1+e^{\frac{nF\Delta E}{2RT}}\right)}C$$
where *i_p_* is the peak current value [A], *C* is the Ferri concentration [mol cm^−3^], *A* is the electrode surface area [cm^2^], *D_eff_* is the effective diffusion coefficient of Ferri [cm^2^ s^−1^], Δ*E* is the pulse height of the input signal [V], *t* is the pulse time [s], *n* is the number of electrons participating in the electrochemical reaction, *R* is the universal gas constant [J K^−1^ mol^−1^], *F* is Faraday constant [A mol^−1^], and *T* is the room temperature [K]. The calculated *D_eff_* value of Ferri for the microfabricated electrode (6.85 × 10^−8^ ± 6.49 × 10^−9^ cm^2^ s^−1^) was two orders of magnitude slower than the one reported in the literature (7.26 × 10^−6^ cm^2^ s^−1^ [35]). The difference in the *D_eff_* values may be due to the difference in the shape and size of the electrodes. Despite this difference, we observed similar electrochemical signal characteristics; therefore, we can assume that the electrochemical activity recorded by the microfabricated electrode had similar Nernstian characteristics.

### 3.2. Characterization of the Chitosan Electrodeposition Process and the Resulting Chitosan-Modified Electrode

We characterized the chitosan electrodeposition rate by measuring the film thickness for different electrodeposition durations. Figure 4A shows the dependence of the chitosan film thickness on the electrodeposition duration, which resulted in a positive linear relationship. The slope of this relationship represented the electrodeposition rate and was calculated as 0.35 ± 0.04 nm s^−1^. While linear dependence was observed for electrodeposition durations between 30–180 s, we expected to observe nonlinear dependence for lower durations due to the stronger hydrolysis effect at the electrode’s surface on the electrodeposition rate.

The electrochemical activity of the chitosan-modified electrode was characterized. Recorded cyclic voltammograms showed increased oxidation and reduction current peaks for the chitosan-modified electrode (24.27 µA and −29.81 µA) in comparison with the bare electrode (20.83 µA and −24.92 µA) (Figure 4B). The increased current peaks observed with the chitosan-modified electrode are due to the negative molecular charge property of the Ferri/Ferro couple that is electrostatically attracted to the positively charged chitosan and is concentrated next to the electrode [36]. Furthermore, we observed lower electrochemical currents from the simultaneously recorded bare working electrodes in the array compared to higher currents for individually recorded electrodes.

### 3.3. Chitosan Film Thickness Affects the Effective Diffusion Coefficients of Redox Molecules with Different Physiological Charges and Molecular Weights

We investigated the effect of the chitosan thickness on the calculated *D_eff_* value for redox molecules of different molecular weights (*M_w_*) and physiological charges (Table 1). The relationships between *D_eff_* and the chitosan thickness for the differentially charged redox molecules are shown in Figure 5 (a list of the calculated *D_eff_* values is presented in Appendix A). For the Ferri molecule (Figure 5A), we observed a piecewise relationship that increased for thinner films (0–30 nm) and decreased for thicker films (30–80 nm). This piecewise relationship suggests that electrostatic attraction between the negatively charged Ferri and the positively charged chitosan occurs for thin chitosan films, whereas, for thicker films, a different mechanism is dominant. For example, thicker chitosan films are denser and can increase the diffusion path of Ferri [37,38], resulting in slower diffusion coefficients. For the AA, no clear relationship was observed between the *D_eff_* and the chitosan thickness (Figure 5B). Therefore, we suggest that the electrostatic attraction forces are negligible between the chitosan film and AA since they are weaker than Ferri due to the lower physiological charge. Moreover, the denser chitosan did not show the clear decreased *D_eff_* effect, which was observed for Ferri due to the lower molecular weight of AA compared with Ferri. We observed for both HCy and CLZ, a negative relationship between *D_eff_* and the chitosan thickness (Figure 5C,D). Since HCy is neutral, no attraction forces are expected between the molecule and the chitosan film. Therefore, we can assume that the dominant mechanism affecting the *D_eff_* value of HCy is the denser chitosan film and the increasing diffusion path. Moreover, since CLZ is positively charged, naturally, we assumed that electrostatic repulsion forces play a dominant role in *D_eff_* changes. However, the increased diffusion path with the denser chitosan film can also affect *D_eff_*; hence, the dominant mechanism remains unknown.

### 3.4. Chitosan Film Can Differentially Affect the Diffusion Rate of Differentially Charged Redox Molecules

We compared the differential effect of the chitosan film thickness on the redox molecules with different physiological charges and *M_w_* properties. To this end, we normalized *D_eff_* values calculated for all the molecules for different chitosan thicknesses (relative *D_eff_* [*r_D_eff_*]; Equation (3)).
*r_D_eff_* = *D_eff_*
_for a specific chitosan thickness_/*D_eff_*
_for a bare electrode_,(3)

The dependence of *r_D_eff_* on the physiological charge and the *M_w_* value of the molecules for different chitosan film thicknesses is presented in Figure 6. For the 33.4 nm-thick chitosan film (Figure 6A), Ferri showed the highest *r_D_eff_* value (1.86 ± 0.67), AA showed a r_*D_eff_* value that was slightly higher than that of the bare electrode (1.08 ± 2.8 × 10^−4^), whereas HCy and CLZ yielded *r_D_eff_* values that are lower than 1 (namely, *D_eff_* values measured using chitosan modified electrodes are lower than their corresponding *D_eff_* values with a bare electrode) (0.54 ± 3.4 × 10^−4^ for HCy and 0.44 ± 2.8 × 10^−4^ for CLZ). For the 40.9 nm-thick chitosan (Figure 6B), Ferri and AA showed similar *r_D_eff_* values that are higher than 1 (1.22 ± 0.21 and 1.18 ± 2.0 × 10^−4^, respectively), whereas HCy and CLZ yielded *r_D_eff_* values that are lower than 1 (0.5 ± 1.9 × 10^−4^ and 0.06 ± 3.6 × 10^−4^, respectively). For the 54.7 nm-thick chitosan (Figure 6C), all the tested molecules showed *r_D_eff_* values lower than 1 (0.85 ± 0.35, 0.90 ± 2.8 × 10^−4^, 0.34 ± 2.7 × 10^−4^, and 0.19 ± 1.3 × 10^−4^ for Ferri, AA, HCy, and CLZ, respectively). Moreover, the *r_D_eff_* values are organized from the highest to the lowest as Ferri and AA > HCy > CLZ. For the 72 nm-thick chitosan (Figure 6D), the *r_D_eff_* value for AA (1.66 ± 7.1 × 10^−4^) was higher than 1 and higher than that of the *r_D_eff_* value of Ferri (0.66 ± 0.25). In addition to the *r_D_eff_* value for Ferri, HCy and CLZ yielded values lower than 1 (0.19 ± 1.4 #xD7; 10^−4^ and 0.14 ± 1.2 × 10^−4^, respectively). For the 84.1 nm-thick chitosan (Figure 6E), all the tested molecules showed *r_D_eff_* values lower than 1 (0.87 ± 0.26, 0.79 ± 7.9 × 10^−4^, 0.26 ± 1.8 × 10^−4^, and 0.14 ± 8.4 × 10^−5^ for Ferri, AA, HCy, and CLZ, respectively). Moreover, the *r_D_eff_* values are ordered from the highest to the lowest as Ferri and AA > HCy > CLZ.

The obtained results show the dominant effect of electrostatic attraction forces between the negatively charged molecules Ferri and AA and the positively charged chitosan. Moreover, the change in the *r_D_eff_* value of Ferri to lower than 1 and with a chitosan thickness of 84.1 nm represents the transition in the dominant effect from the electrostatic force to the longer diffusion length. However, this transition in the effects on AA was observed with a thinner chitosan film (54.7 nm). Such a difference may be related to the physiological charge of Ferri, which is higher than for AA (−3 vs. −1); it requires thicker films to decrease the influence of the magnitude of the electrostatic attraction forces. The neutral molecule HCy resulted in *r_D_eff_* values lower than 1 for all chitosan films. These results support our assumption in Section 3.3 that the main mechanism affecting the diffusion rate of HCy is the denser chitosan and the longer diffusion paths. CLZ decreased the *r_D_eff_* values for all chitosan films; this can be attributed to two suggested mechanisms: electrostatic repulsion and a longer diffusion length. Both mechanisms can affect the CLZ diffusion rate because no transition in the trend was observed from the obtained results. Therefore, a better differentiation mechanism can be achieved by further investigating the diffusion of CLZ in the chitosan film with direct methods.

### 3.5. Simultaneous Prediction of Molecules’ Concentration from Mixture Samples

We tested the ability to differentiate between overlapping electrochemical signals and quantify redox-active molecules from a multicomponent mixture. To this end, we used an array of electrodes modified with different chitosan thicknesses to record a set of diffusion-influenced complex electrochemical signals; we utilized the chemometric model PLSR to analyze the signals. As input variables, we used the whole electrochemical spectrum (differential pulse voltammogram; current vs. potential). To train the model, we recorded signals for mixtures containing four redox-active molecules at four different concentrations (UA: 140, 260, 380, and 500 µM; AA: 0, 200, 500 and 1000 µM; HCy: 150, 300, 400, and 500 µM; CLZ: 1, 2.5, 3.75, and 5 µM; a total of 64 solutions. The compositions of the mixtures are shown in Appendix A). Appendix A shows the electrochemical signatures measured from the same concentration of AA (black), UA (red), CLZ (blue), and HCy (magenta) by using the gold bare electrode. Appendix A shows the complex electrochemical voltammograms recorded from the multicomponent mixture solutions (prepared according to Appendix A) using the bare and the chitosan-modified electrodes. The trained model was tested with an additional set of signals measured from 31 mixtures with different concentrations of UA, AA, HCy, and CLZ. (The compositions of the mixtures are shown in Appendix A) The PLSR model (leave-one-out cross-validation resulted optimal 12 latent variables; 97.1% cumulative X variance and 98.8% cumulative Y variance (Appendix A)) predicted three out of the four redox-active molecules (Figure 7); AA, UA, and HCy were successfully predicted with a root mean square error (RMSE) (Appendix A), PCC, and coefficient of determination (R-squared) values of 125 µM, 0.947, and 0.894; 51.8 µM, 0.877, and 0.753; 55.7 µM, 0.903, and 0.809, respectively (Table 2). To improve CLZ prediction performance, nonlinear models (such as the artificial neural network) can explain the high complexity of the electrochemical signals generated from the mixture. Furthermore, a small dataset can lead to models with high prediction errors. Therefore, a statistically accurate model can be achieved by overcoming technological barriers that will enable generating a large dataset of electrochemical signals.

## 4. Discussion and Conclusions

This work used a novel approach to differentiate between multiple redox-active molecules with overlapping electrochemical signals. An array of electrodes was fabricated and modified with different thicknesses of the positively charged chitosan to differentially affect the diffusion rates of redox-active molecules with different physiological charges. The modified array generated a complex set of electrochemical signals to train a chemometric PLSR model. The trained model successfully predicted three out of four redox-active molecules in multicomponent mixtures.

Although the influence of the chitosan film on the diffusion rates can be due to repulsion-attraction electrostatic forces and longer diffusion paths based on the density of the chitosan, further enhancing our understanding of the dominant influential mechanism will reveal additional features in the electrochemical signals that will increase the differentiation resolution between redox-active molecules with similar standard reduction potentials. The successful proof-of-concept study presented here can be used to overcome the main challenge of electrochemical sensors for in situ analysis of biofluids—simultaneous quantification of multiple redox-active biomarkers. Whereas here, we imply that this work aims to perform the analysis for biomedical applications. We also suggest that this analytical approach can also be used for other applications, such as the food and environmental monitoring industries.

## Figures and Tables

**Figure 1 polymers-14-00717-f001:**
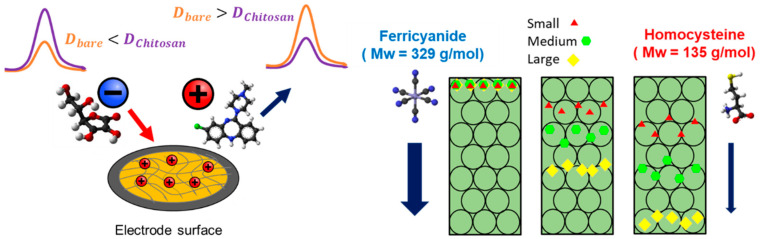
Scheme describing the differentiation of redox molecules by using diffusion-limiting chitosan-modified electrodes. Since the chitosan film is positively charged, the negatively charged redox molecules are electrostatically attracted to the electrode, resulting in faster diffusion rates. Moreover, for chitosan films of higher densities, redox molecules with smaller molecular weights can undergo a longer diffusion path resulting in a slower diffusion rate toward the electrode.

**Figure 2 polymers-14-00717-f002:**
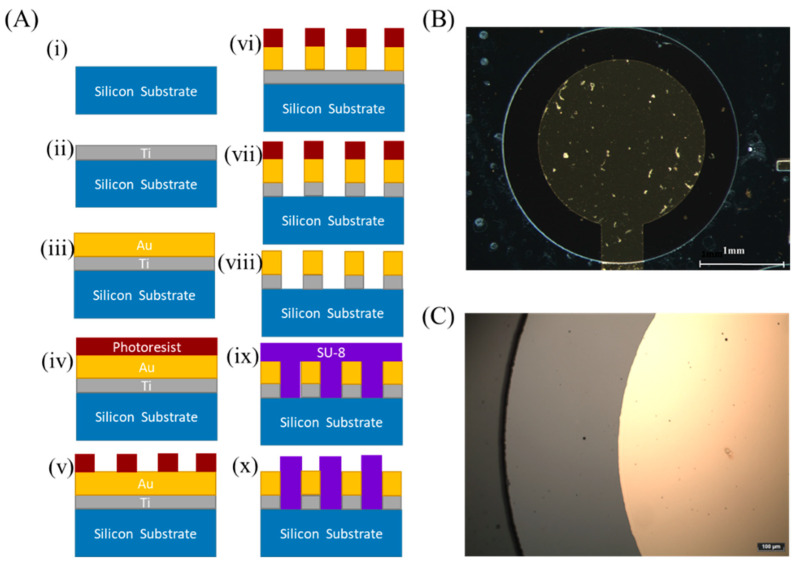
Microfabrication of the gold electrodes. (**A**) A microfabrication flow process: (**i**) Si/SiO_2_ substrate was cleaned, (**ii**) Ti was deposited onto the Si/SiO_2_ wafer, (**iii**) Au was deposited onto the Si/SiO_2_/Ti, (**iv**) a TiX-Lift photoresist was spin-coated onto the wafer, (**v**) the photoresist was patterned with UV and developed, (**vi**) Au was etched from the exposed areas, (**vii**) Ti was etched from the exposed areas, (**viii**) a photoresist was removed from the wafer, (**ix**) a SU-8 photoresist was spin-coated on the wafer patterned with electrodes, and (**x**) the photoresist was patterned with UV and developed to pattern the chambers. (**B**) An optical image of the whole microfabricated electrochemical chamber patterned with SU-8. (**C**) A zoomed-in image of the microfabricated electrode with the SU-8 chamber.

**Figure 3 polymers-14-00717-f003:**
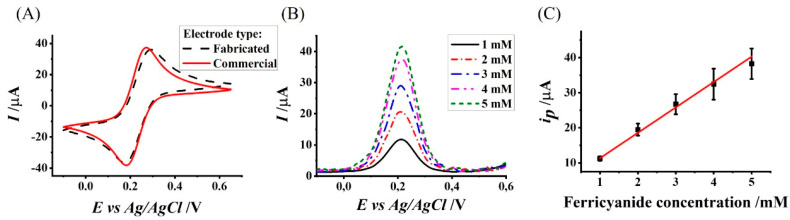
Electrochemical characterization of the microfabricated electrodes. (**A**) Cyclic voltammograms recorded using a commercial (solid red) and the microfabricated (dashed black) electrodes in a 5 mM Ferri/Ferro solution. (**B**) Voltammograms recorded in the presence of 1, 2, 3, 4, and 5 mM Ferri concentrations. (**C**) A calibration curve plot of the oxidation peak current as a function of the Ferri concentration recorded using the microfabricated electrode.

**Figure 4 polymers-14-00717-f004:**
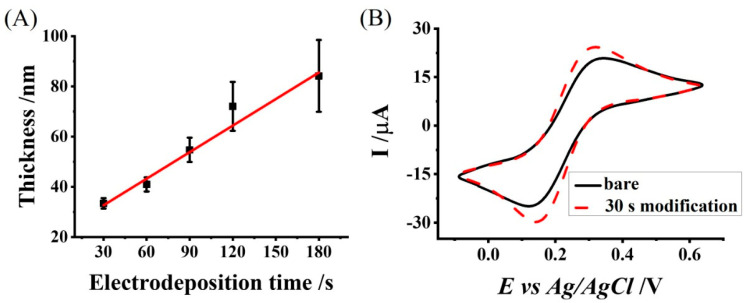
Chitosan-modified electrode characterization. (**A**) The thickness of the chitosan layer for increasing electrodeposition durations; (**B**) Cyclic voltammograms recorded with the bare (solid black) and the chitosan-modified (30 s electrodeposition duration; dashed red) electrodes.

**Figure 5 polymers-14-00717-f005:**
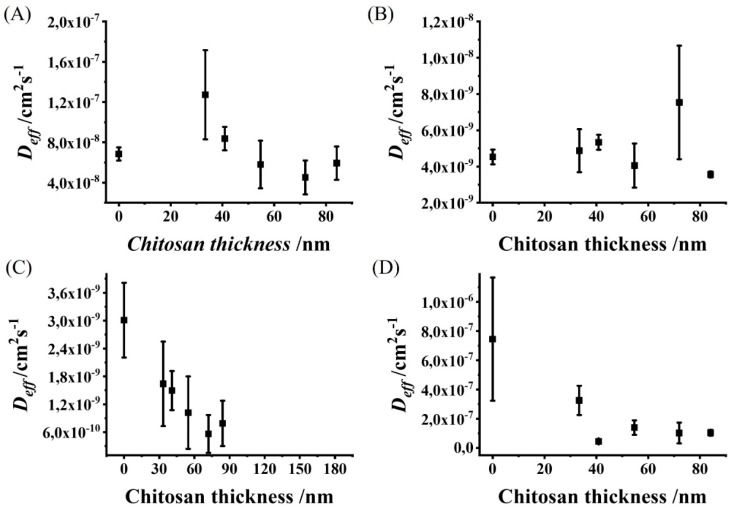
The influence of the chitosan thickness on the diffusion rate for differentially charged redox-active molecules. *D_eff_* dependence on the chitosan film thickness for (**A**) Ferri, (**B**) AA, (**C**) HCy, and (**D**) CLZ redox-active molecules.

**Figure 6 polymers-14-00717-f006:**
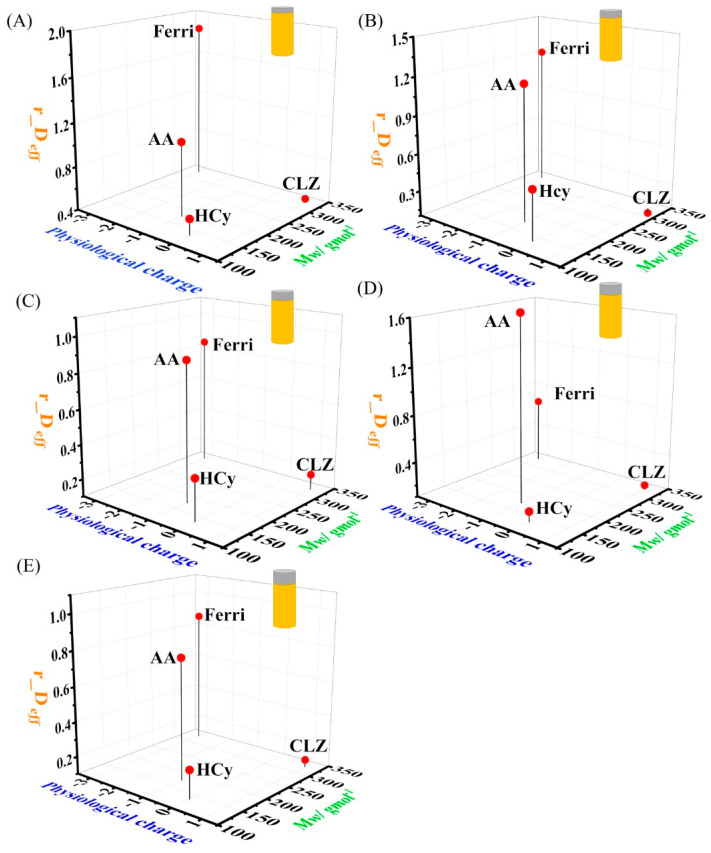
The differential influence of the physiological charge and the molecular weight of the redox-active molecules on the diffusion rate. The dependence of *r_D_eff_* on the *M_w_* value and the physiological charge measured for Ferri, AA, HCy, and CLZ and for a chitosan thickness of (**A**) 33.4 nm, (**B**) 40.9 nm, (**C**) 54.7 nm, (**D**) 72 nm, and (**E**) 84.1 nm.

**Figure 7 polymers-14-00717-f007:**
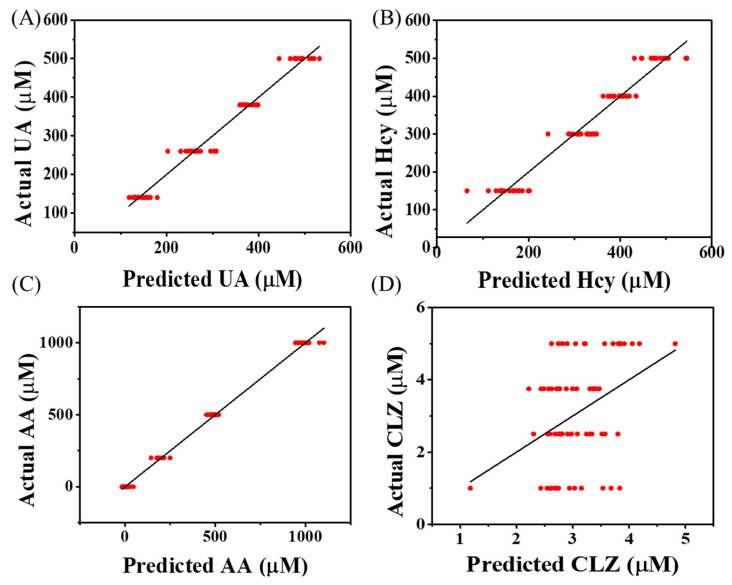
Diffusion- and chemometric-based multi-electrode array to quantify multiple redox-active molecules from multicomponent mixtures. Predicted vs. actual concentrations for (**A**) UA, (**B**) HCy, (**C**) AA, and (**D**) CLZ, calculated from the chemometric-based multi-electrode array.

**Table 1 polymers-14-00717-t001:** Molecular weight and the physiological charge properties of the tested redox-active molecules.

	Ferri	AA	HCy	CLZ
*Mw* [g mol^−1^]	329.24	176.12	135.18	326.82
Physiological charge	−3 [39]	−1 [39]	0 [40]	+1 [41]

**Table 2 polymers-14-00717-t002:** Chemometric model prediction performance.

	AA	CLZ	UA	HCy
RMSE (µM)	125	1.63	51.8	55.7
PCC	0.947	−0.093	0.877	0.903
R^2^	0.894	0	0.753	0.809

## Data Availability

The data presented in this study are available on request from the corresponding author.

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
