# Peer review of "Diffusion- and Chemometric-Based Separation of Complex Electrochemical Signals That Originated from Multiple Redox-Active Molecules"

_polymers, 2022, doi:10.3390/polym14040717_

Round 1

Reviewer 1 Report

The paper looks at detecting four redox-active biomarkers with different thicknesses of chitosan modified electrodes. This is an interesting study to alter the diffusion of these biomarkers. However, several improvements can be made to the paper to improve the impact and clarity.

Figure 6, and the discussion around Figure 6, is poorly explained. The figure itself is unclear on how/why the Molecular weight of each biomarker is being altered from A-E (look at HCy and AA). Also, it looks like the charging is changing on several species. The text doesn't explain how to read or understand this complicated 4-D image. 

Supplemental figures, and several supplemental tables, are not explained in the text of the paper. Specifically Figures S1 and S2 are extremely important and needs a better description in the main text.

What variable was used to input into the PLSR model? (E.g. was it the peak current, current at a specific voltage, total current, etc?) It seems like picking the peak current (without a model) should be able to separate enough of the analytes except CLZ, which is what the "model" predicts. This can be seen in the horizontal lines in Figure 7 which is likely to be due to competing analyte interference. Therefore, greater clarity is needed on what work the model is doing in its predictive analysis. 

The concluding statement, "The successful proof-of-concept study presented here can be used to overcome the main challenge of electrochemical sensors for in situ analysis of biofluids simultaneous quantification of multiple redox-active biomarkers" does not seem well supported. If the authors are implying by adjusting the thickness to get a different effective diffusion coefficient (Table S1), that isn't supportive as it appears thickness is independent on effective diffusion (again see Table S1). If the authors are implying the PLSR model, please see comments above. Greater clarity of the data is needed to support this claim and other concluding claims.  

Reviewer 2 Report

The paper discusses the potential of using a chitosan film to facilitate the differentiation between different redox active moieties in the same solution. It does this by a determining the diffusion constant of the redox moiety and links the changes mainly to the charge of the moiety.

This raises my main concern about the paper. The authors already state that the diffusion of the redox molecules through the chitosan film is determined by the physiological charge as well as the size of the molecule for which we can take the molecular weight as a proxy. However, the authors go into an extensive analysis of the correlation between the charge and the diffusion constant, while at the same time completely ignoring the molecular weight as a potential factor. The only comparisons that can be made is between the Ferro/Ferri couple and the CLZ, or AA and HCy, as each of these combination have similar molecular weight.

There are a couple of other scientific questions that I have after reading the paper:

  • In lines 180 and 181, the authors list the range of concentrations of the various redox molecules that they will be using. What is the justification for using these specific concentrations?
  • In line 211, the authors make an assumption that the diffusion constant on the fabricated electrodes is the same to that measured on the commercial electrodes. As this is a fairly straightforward experiment, why was this not experimentally verified?
  • In lines 193 and 229, the authors measure the current of the oxidation and reduction peak of the bare electrode with as far as can be deducted from the paper, the same conditions, why then is there a difference of 33% (~10 µA) between the two experiments?
  • In Figure 4A. the authors show the chitosan thickness as a function of the deposition time. I am fairly certain that if the deposition time is 0 the thickness is 0, however, in Figure 4A. it looks like there is a layer of 20 nm thick for 0 s of deposition?
  • The paragraph/ sentences describing the results (280 till 293) appear to be a repetition of the data presented in Figure 6. Neither the text nor the figure actually helps the audience to understand the results. I would suggest the authors consider an alternative way of presenting the information.
  • The machine learning protocol clearly does not work for CLZ, while the authors do state this, they do not provide a discussion as to why this might be the case.

A couple of more textual comments:

  • The authors are inconsistent in their use of spaces between value and unit.
  • The way the equations are presented within the paper is not beneficial for the reader to understand them.
  • The figures in Figure 5 are too small.
  • The term rDeff is confusing as it is too close to the Deff.

Round 2

Reviewer 1 Report

The author addressed all initial comments sufficiently and thoroughly. No other changes needed. 

Reviewer 2 Report

I am happy with the changes the authors have made.

I still believe that the comparison between the commercial and fabricated electrodes should have been measured rather than presumed to be identical.